# Systemic Inflammation and Lung Cancer: Is It a Real Paradigm? Prognostic Value of Inflammatory Indexes in Patients with Resected Non-Small-Cell Lung Cancer

**DOI:** 10.3390/cancers15061854

**Published:** 2023-03-20

**Authors:** Antonio Mazzella, Elena Maiolino, Patrick Maisonneuve, Mauro Loi, Marco Alifano

**Affiliations:** 1Thoracic Surgical Department, Cochin Academic Hospital, APHP, Université de Paris, 75014 Paris, France; 2Division of Epidemiology and Biostatistics, IEO, European Institute of Oncology IRCCS, 20141 Milan, Italy

**Keywords:** lung cancer, inflammation, HALP, NSCLC, inflammatory status

## Abstract

**Simple Summary:**

Systemic inflammation and changes in the inflammatory status are frequent features of lung cancer. There is a close interconnection between cancer development and the clinical, general, and inflammatory status of patients. In this paper, we evaluate a large panel of inflammatory indexes in patients who underwent lung resection for NSCLC lung cancer; we show that pre-operative inflammatory status strongly influences long-term prognosis in patients affected by NSCLC and undergoing surgery.

**Abstract:**

**Background (1):** Our goal was to investigate if and how pre-operative inflammatory status can influence the long-term prognosis of patients undergoing lung surgery for cancer. **Materials and Methods (2):** This prospective observational study includes the agreement of all operable patients to the study, who were referred to our department between 1 January 2017 and 30 December 2018. The inflammatory pre-operative status of the patients was investigated by calculating albumin, CPR (c-protein reactive), complete blood count (neutrophils, lymphocytes, platelets, hemoglobin), and some other indexes referring to inflammatory status, namely the HALP amalgamated index, platelet-to-lymphocyte ratio (PLR), neutrophil-to-lymphocytes ratio (NLR), systemic immune-inflammation index (SII), and advanced lung cancer inflammation Index (ALI). The follow-up ended in November 2021. Patient overall survival was assessed using the Kaplan–Meier method. The log-rank test was used to compare survival rates. Variables significantly associated with survival at univariate analysis were entered int Cox multivariate analysis (stepwise mode) to assess their independent character. Hazard ratios and their 95% confidence intervals were calculated. Variables associated with *p* < 0.05 were considered significative. **Results (3)**: We enrolled 257 patients in our study. The overall survival of the cohort was as follows: 1 year, 96.1%; 3 year, 81.3%; and 4 year, 74.2%. Univariate analysis showed risk factors for overall survival as follows: Thoracoscore ≥ 2 (*p* = 0.002); histology (*p* = 0.002); HALP < 32.2 (*p* = 0.0002); SII ≥ 808.9 (*p* = 0.0004); ALI < 34.86 (*p* = 0.0005); NLr ≥ 2.29 (*p* = 0.01); hemoglobin < 13 g/dl (*p* = 0.01); PLR ≥ 196.1 (*p* = 0.005); pN+ (*p* < 0.0001); pleural invasion (*p* = 0.0002); and presence of vascular or lymphatic tumor emboli (*p* = 0.0002). Multivariate Cox analysis (stepwise model) identified Thoracoscore ≥ 2 (*p* = 0.02); histology, HALP < 32.2 (*p* = 0.004), and pN (*p* < 0.0001) as independent predictors of death. **Conclusion (4):** Pre-operative inflammatory status strongly influences long-term prognosis in patients affected by NSCLC and undergoing surgery.

## 1. Introduction

Lung cancer is the second most frequent cancer in both sexes, but it still represents the leading cause of cancer-related death in males and the second in females [1]. The largest part of primary malignancy of the lung comprises non-small-cell lung cancer (NSCLC); small-cell lung cancer (SCLC) is less common, representing between 10–15% of all primary lung cancers [2].

Systemic inflammation and impairment of nutritional status are frequent features of lung cancer [3,4]. These two factors are strongly correlated in patients undergoing surgery for lung cancer, and both are related to the limited infiltration of the tumor by immune cells [4]. Inflammatory status (pre-existent or concomitant with lung cancer) is related to subsequent increased energy consumption and contributes to malnutrition, catabolic processes secondary to inflammation, reduced caloric intake, and an imbalance in anabolic and proteolytic pathways; these aspects may be responsible for fat and muscle loss and cachexia, feeding inflammatory status, and causing a vicious cycle of these side effects [5,6].

There is, in fact, a close interconnection between cancer development and clinical, general, and inflammatory status. Cancer arises more easily in chronically inflamed tissue [7]. Inflammatory molecules may be responsible for augmented macrophage recruitment, delayed neutrophil clearance, and an increase in reactive oxygen species. Chronic pulmonary disorders cause an increased release of cytokines and growth factors that represent the cornerstone for epithelial-to-mesenchymal transition and for the constitution of the tumor microenvironment. Many papers have demonstrated the close link between chronic inflammation/lung cancer and between cancer-related inflammation and shorter survival [8,9,10,11,12,13,14,15,16,17,18,19,20].

Our goal was to investigate if and how pre-operative inflammatory status influences the long-term prognosis of patients undergoing lung surgery for cancer. In particular, we focused not only on the role of each individual inflammatory parameter, but also on other complex scores in order to evaluate if single parameters, alone or in association with each other, can really influence prognosis.

## 2. Materials and Methods

This was a prospective observational study accepted by the Ethics committee (n. CERC-SFCTCV-2017-1-8-20-26-9-AlMa). We analyzed the long-term outcomes of the first part (257 patients) of a 1077 patient series, conducted between 2017 and 2021 in our department [21] (Figure 1). The period of patient inclusion started on 1 January 2017 and ended on 30 December 2018. Our study included all operable patients who agreed to the study, were referred for thoracic surgery at Cochin Hospital (APHP, Assistance Publique Hôpitaux de Paris), and affected by lung cancer. The research was conducted according to STROBE (Strengthening the Reporting of Observational Studies in Epidemiology [22]) and TRIPOD (Transparent Reporting of a Multivariable Prediction Model for Individual Prognosis or Diagnosis [23]) statements.

### 2.1. Inclusion Criteria

All patients underwent anatomic lung surgery (pneumonectomy, lobectomy/anatomic segmentectomy, wedge resections) for lung cancer, specifically the following patients:-patients in stage I–II (T1-3/N0-1);-patients in stage III (T1-2/N2, single-positive mediastinal lymph node station), treated by chemotherapy or radiotherapy before surgery.

### 2.2. Exclusion Criteria

Patients with contralateral mediastinal lymph nodes (N3), patients with chest wall or mediastinal involvement and mediastinal positive lymph nodes (stage IIIB), and metastatic patients (stage IV a, b, c) were excluded from this study. We also excluded all patients undergoing surgery for SCLC and typical and atypical carcinoids, patients enrolled after 2018 for a shorter follow-up, and patients with incomplete pre-operative biological panels or with a panel performed outside of our departmental laboratory.

### 2.3. Patient Inclusion and Pre-Operative Collection Data

All patients provided written informed consent during the first consultation to participate in the study. Before intervention, all patients underwent routine examinations, such as chest roentgenogram, fiberoptic bronchoscopy for mediastinal staging, contrast-enhanced thoracic and upper abdominal computed-tomography scan, and cerebral magnetic resonance imaging. Positron emission tomography scans were routinely performed. Mediastinal nodes were considered negative at clinical staging if the short axis was less than 1 cm and there was no significant (standardized uptake value < 2.5) [18F] fluoro-2-deoxy-2-D-glucose uptake. In patients with proven pN2 involvement, lung resection was not performed. Functional assessment included spirometry and perfusion lung scan, yielding calculations of predicted post-operative forced-expiratory volume in 1 s. Patients with predicted post-operative forced-expiratory volume in 1 s exceeding 40% were considered operable. Diffusion capacity of the lung for carbon monoxide and 6 min walking test were taken into account. All patients underwent a pre-operative echocardiography with assessment of the left-ventricular ejection fraction and an assessment of systolic pulmonary pressure. For all patients, we calculated Thoracoscore at the time of hospitalization [24] (Table 1). Thoracoscore was described in 2007 as a model for the risk of in-hospital death among adult patients after general thoracic surgery. It uses only nine pre-operative variables (age, sex, ASA classification, performance status, dyspnea score, priority of surgery, procedure, diagnosis, comorbidity) and is recognized as a valid clinical tool for predicting the risk of death. Thoracoscore evaluation is easily calculated and is available online (https://sfar.org/scores2/thoracoscore2.php, accessed on 16 March 2023).

### 2.4. Post–Operative Collected Data and Follow-Up

During hospitalization, treatment procedures and short-term outcomes were collected using a standardized case report form. Data included use of induction chemotherapy, type of lung resection, histologic type, main diameter of tumor, and presence of lymphovascular emboli or perineural spreading. The presence of vascular or lymphatic tumor emboli was assessed by standard hematoxylin and eosin staining on samples from tumor tissue and adjacent non-tumoral lung tissue and defined as the presence of aggregates of tumor cells inside vascular or lymphatic microvessels. Tumor stage was reattributed according to the *TNM Classification of Malignant Tumours, 8th Edition* (Table 2).

After surgery and discharge from the hospital, patients were reviewed for the first time after one month in a routine post-operative consultation. Follow-up information was obtained by telephone interview with patients or by consulting municipality registers. In some cases, we consulted the online form of the French registry office (https://arbre.app/insee/, accessed on 16 March 2023).

### 2.5. Indexes of Inflammatory Status

Inflammatory pre-operative status of the patients was investigated by the analysis of various parameters, including albumin, prealbumin, CPR (c-protein reactive), and complete blood count (neutrophils, lymphocytes, platelets, hemoglobin) (Table 2). Starting with the blood count measurements, we calculated different indexes referring to inflammatory status (Table 3) as follows:-platelet-to-lymphocyte ratio (PLR) and albumin multiplying lymphocytes known as the prognostic nutritional index (PNI);-HALP amalgamated index, which is measured as hemoglobin (g/L) x albumin (g/L) x lymphocyte (/L)/platelet (/L);-serum polymorpho-nuclear neutrophil-to-lymphocyte ratio (NLR);-systemic immune-inflammation index (SII): serum platelets * neutrophil/lymphocytes;-advanced lung cancer inflammation index (ALI): serum albumin * BMI/NLR; BMI = weight (kg)/height (m)^2^.

### 2.6. Data Analysis, Follow-Up, and Statistical Analysis

The results are expressed as the percentage for qualitative parameters and mean SD for quantitative variables. Optimal cut-offs for continuous biological variables were determined by Youden’s J statistic using receiver operating curve (ROC) analysis (Figure 2). Different cut-offs are shown in the table, which are precisely as follows: HALP, 32.2; NLR, 0.01; PLR, 196.1; SII, 808.9; and ALI, 34.86. Overall survival was calculated from the date of surgery to the date of death, or the date of last contact with the patient. Overall survival curves were plotted using the Kaplan–Meier method. The log-rank test was used to assess differences in OS between groups. Hazard ratios (HRs) with 95% confidence intervals (CIs) were calculated using the univariate Cox regression model. Multivariate stepwise regression was then used to identify independent factors associated with OS. *p*-values were two-sided and those <0.05 were considered significant. All analyses were performed with SAS software (version 9.4, Cary, NC, USA).

## 3. Results

We enrolled 257 patients in our study (149 males, 108 females). The mean age was 65 years (DS 10.2). Clinical and biological data are represented in Table 1.

### 3.1. Pathological, Functional, and Biological Findings

The pathological post-operative findings revealed the following rates: adenocarcinoma (188 patients, 73.2%); squamous-cell carcinoma (54 patients, 21.0%); and other NSCLC (15 patients, 5.8%).

According to the eighth TNM, we found 145 patients with stage I (56.4%) cancer, 57 patients with stage II (22.2%) cancer, and 55 patients with stage III (21.4%) cancer. The post-operative lymph node stage revealed pN0 in 187 patients (72.8%), pN1 in 33 patients (12.8%), and pN2 in 33 patients (12.8%).

Analysis of pleural invasion on surgical specimen showed Pl0 in 164 patients (63.8%) and Pl+ in 93 patients (36.2%).

In 102 cases (39.7%), no perineural and/or vascular neoplastic emboli were found in the specimen. In 154 (59.9%) cases, we found neoplastic perivascular emboli.

All functional and metabolic findings are reported in the table.

### 3.2. Thirty-Day Mortality and Post-Operative Complications

No deaths 30 days after surgery were observed. One hundred and thirty-five patients presented minor and major complications. In particular, we observed prolonged air leaks in 80 patients, post-operative hemothorax necessitating surgical revision in 8 patients, pneumonia necessitating broncho-aspiration and NIV (non-invasive ventilation) in 7 patients, respiratory failure necessitating ICU stay in 4 patients, multi-organ failure in 2 patients, and atrial fibrillation in 34 patients.

### 3.3. Overall Survival, Univariate, and Multivariate Analysis

During a median observation time of 40 months (interquartile range, 33–46), 56 of the 257 patients died. The average annual death rate was 6.9% (56/806 patient years of observation). The overall survival of the cohort was as follows: 1 year, 96.1%; 3 year, 81.3%; and 4 year, 74.2% (95% CI, 67.3–79.9) (Figure 3).

Univariate analysis showed the risk factors for overall survival as follows: Thoracoscore ≥ 2 (*p* = 0.002); histology (*p* = 0.002); HALP < 32.2 (*p* = 0.0002); SII ≥ 808.9 (*p* = 0.0004); ALI < 34.86 (*p* = 0.0005); NLr ≥ 2.29 (*p* = 0.01); hemoglobin < 13 g/dl (*p* = 0.01); PLR ≥ 196.1 (*p* = 0.005); pN+ (*p* < 0.0001); pleural invasion (*p* = 0.0002); and presence of vascular or lymphatic tumor emboli (*p* = 0.0002) (Table 1, Table 2 and Table 3). Multivariate Cox analysis (stepwise mode) identified as risk factors: Thoracoscore ≥ 2 (*p* = 0.02); HALP < 32.2 (*p* = 0.004); and histology and pN+ (*p* < 0.0001) (Table 4, Figure 3).

Based on univariate and multivariate analysis, we calculated correlations between the level of inflammatory markers and the status of lymph node involvement or disease stage. HALP (*p*: 0.008), PLR (0.004), SII (<0.0001), and ALI (0.02) were associated with the more invasive stages (Table 5).

## 4. Discussion

A plethora of studies have demonstrated that there is a close interconnection between inflammation and poor prognosis of cancer [8,9,10,11,12,13,14,15,16,17,18,19,20].

The most common abnormalities, linked to chronic inflammatory processes accompanying neoplasia, are represented by leukocytosis, neutrophilia, thrombocytopenia, and lymphocytopenia. These may occur during the growth and lysis of the tumor [17,25,26]. The positive feedback of immunoregulatory cytokines in the recruitment of tumor-associated neutrophils induces disease progression and increases the risk of distant metastasis; platelets also appear to have a similar role. In contrast, lymphocytes are believed to have anticancer activity; indeed, lymphocytosis itself is considered a favorable prognostic factor [27]. Platelet count was found to be increased in lung cancer and colorectal cancer, which indicated poor survival outcomes [9]. Lymphocytes play an important role in defense against cancer by inducing cytotoxic cell death and inhibiting proliferation and migration of cancer cells [10]. Hemoglobin has been reported to be a prognostic factor in cancer patients, and anemia is associated with poor prognosis [11]. Albumin has also been demonstrated as a prognostic factor in gastric cancer, revealing that patients with higher levels of albumin had better prognosis than those with lower levels of albumin [12].

Changes in tumor-related inflammatory cells are strictly linked to the degree of inflammatory response to tumors; indeed, a higher inflammatory response often indicates a worse prognosis. On one hand, high levels of hemoglobin, albumin, and lymphocytes may be positively correlated with prognosis; on the other hand, high levels of platelets may be associated with poor prognosis.

In wider terms, inflammatory status impacts the quality of life of the patients in terms of their immune response to cancer, particularly on the metabolism of lung cancer and of the host. It is well known that there is a close connection between the inflammatory status and metabolic status of these patients. On one hand, there is the absence of body reserves occurring in some lung cancers, inducing cachexia or a catabolic state favoring cancer development; this condition (up to malnutrition) is the cause of immunodeficiency promoting cancer progression. Thus, this cancer-related condition could be responsible for immunity dysregulation, catabolic state, and subsequent promotion of cancer itself. In addition, host metabolism is altered in order to produce more glucose, required by cancer cell proliferation; anaerobic glycolysis could therefore be preferred, i.e., the Warburg effect, and this aspect could be associated with poor differentiation and survival [28,29,30]. Lactates resulting from anaerobic glycolysis (the Warburg effect) could alter immune response because they influence the uptake of glucose of cytotoxic cells in the tumoral microenvironment. This effect could be a downregulation of CD8+ cytotoxic T-cells in the local control of cancer progression.

In our study, we investigated not only the role of each individual inflammatory parameter, but also the other complex scores in order to evaluate whether single parameters, alone or in association, could really influence long-term prognosis. Indeed, further research has shown that a combination of these parameters can better predict a patient’s prognosis than a single index. Among them, HALP score, calculated as Hemoglobin (g/L) × Albumin (g/L) × Lymphocyte (/L)/Platelet (/L), was reported to be related to survival in gastric cancer, colorectal cancer, bladder cancer, and renal cancer patients [18,19,20]. Chen et al. demonstrated that the HALP index was associated with tumor size and T stage. Low HALP was significantly associated with tumor progression and acted as an adverse prognostic factor in gastric cancer [31] and lung cancer [18,19,20]. The neutrophil–lymphocyte ratio (NLR) was reported as a novel serum inflammatory marker in patients with advanced or recurrent NSCLC, treated with molecular targeted therapy or immunotherapy [32,33,34,35,36,37,38]. The authors reported the important role of cancer-specific cytotoxic T-cells for anticancer response and the function of tumor-associated macrophages to promote tumor angiogenesis. Choi et al. [14] demonstrated that a high pre-operative NLR (≥5), a marker of inflammation, is associated with a decrease in RFS and OS in patients with early-stage NSCLC. These results are in agreement with those reported by Forget [13] and Sarraf [16] who demonstrated that, in a much smaller population of patients with stage I and II NSCLC, a pre-operative NLR ≥ 5 was an independent risk factor for worse RFS and OS [16,17]. The platelet-to-lymphocytes ratio (PLR) and albumin multiplying lymphocytes, known as the prognostic nutritional index (PNI), have been extensively studied in lung cancer [13]. Lokowki et al. [17] concluded that in NSCLC patients, elevated PLR (platelet–lymphocyte ratio) values appear to be an independent prognostic factor for survival. We considered other indexes, namely SII (systemic immune-inflammation index) or ALI (advanced lung cancer inflammation Index). Fournel [39] showed the strong impact of systemic inflammation on the prognosis of malignant pleural mesothelioma and a shorter survival associated with NLR, SII, and lower ALI.

We can therefore assert that in patients affected by lung cancer, there is important reorganization in the immune system, particularly an increase in platelets and neutrophils; contrarily, lymphocytes tend to decrease with a subversion of the different inflammatory parameters, such as NLR, PLR, HALP, SII, and ALI.

In our work, the strong association between the different inflammatory indexes analyzed and poor prognosis clearly corroborates the other literature data. In addition, we analyzed and correlated, for the first time, some other parameters, such as ALI, SII, and HALP, with the long-term prognosis of resected lung cancer.

The other parameters impacting survival were the oncologic characteristics of the tumor (stage and N status) and the presence of vascular or lymphatic tumor emboli. In the multivariate analysis, lymph node involvement strongly influences overall survival. Lymph node status is universally recognized as one of the main prognostic factors in lung cancer survival, both in locally advanced, oligo-metastatic, or metastatic disease [40,41,42,43]. Indeed, the presence of lymph node metastases (N1–N2) is a direct expression of aggressiveness in the neoplasm and of its propensity for recurrence or distant spread.

Another variable associated with longer overall survival at univariate analysis was the absence of lymphovascular and perineural neoplastic emboli in the lung tumor specimen. We can consider the presence of the emboli as an indicator of aggressiveness, considered the first step to lymphnodal or systemic spread, or even an indicator of occult metastases in early stages of disease [44]. Our results are in accordance with other studies [43,44].

Finally, the last outcome considered to be risk factor for OS was Thoracoscore. This is a validated multivariate model for risk of in-hospital death among adult patients after general thoracic surgery described for the first time in 2007 [24]. It includes nine variables as follows: age; sex; ASA classification; performance status classification; dyspnea score; priority of surger; procedure class (pneumonectomy vs no pneumonectomy); diagnosis group (benign or malign); and comorbidity score (0–2, =2, >2). Our results corroborate the results of other reports [24,45], in which Thoracoscore is considered a good and useful clinical tool for the prediction of pre-operative and midterm mortality in operated patients.

Our study presents some limitations. For example, the prospective nature of the study allowed us to have a homogeneous cohort, with particularly detailed outcomes. Even if our sample is large enough (257 patients), our conclusions should be corroborated by a larger-scale evaluation. Another limitation could be the correlation between different parameters (albumin, hemoglobin, HALP, SII, or blood count cell), impacting survival at univariate and multivariate analysis; in order to limit this bias and reinforce our results, we calculated HRs and 95% ICs for each outcome, confirming their negative impact on survival.

## 5. Conclusions

The prognosis of NCLSC is influenced by systemic inflammation. Alteration of pre-operative inflammation indexes has an important impact on survival, and this aspect generally corroborates the concept of a close correlation between systemic inflammation and long-term prognosis in NSCLC. Other studies are mandatory to validate and corroborate our results.

## Figures and Tables

**Figure 1 cancers-15-01854-f001:**
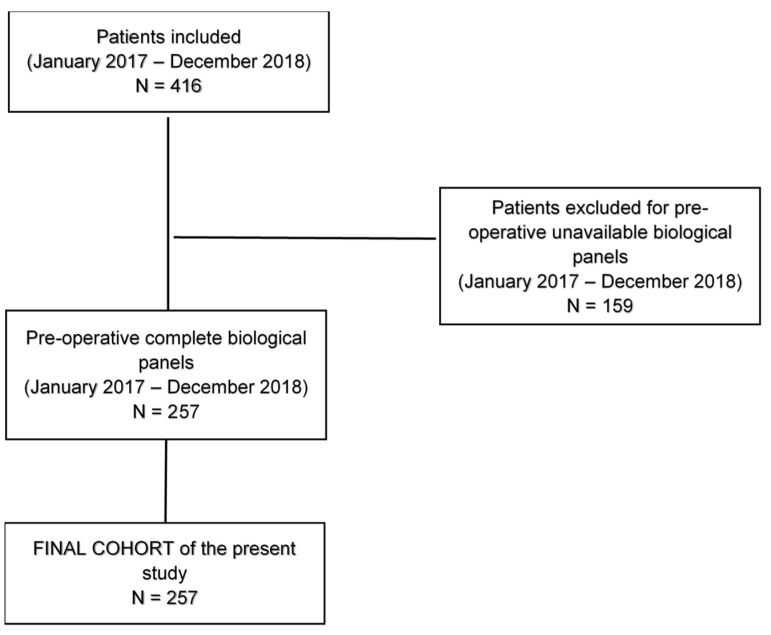
Flow chart of the study. In this study, we analyzed the long-term outcomes of the first part (257 patients) of a 1077 patient series (January 2017–August 2021) [21], undergoing anatomical surgical resection for non-small-cell Lung Cancer. We included all patients undergoing anatomic lung resection in the period January 2017–December 2018, with pre-operative available biological panels performed in our departmental laboratory.

**Figure 2 cancers-15-01854-f002:**
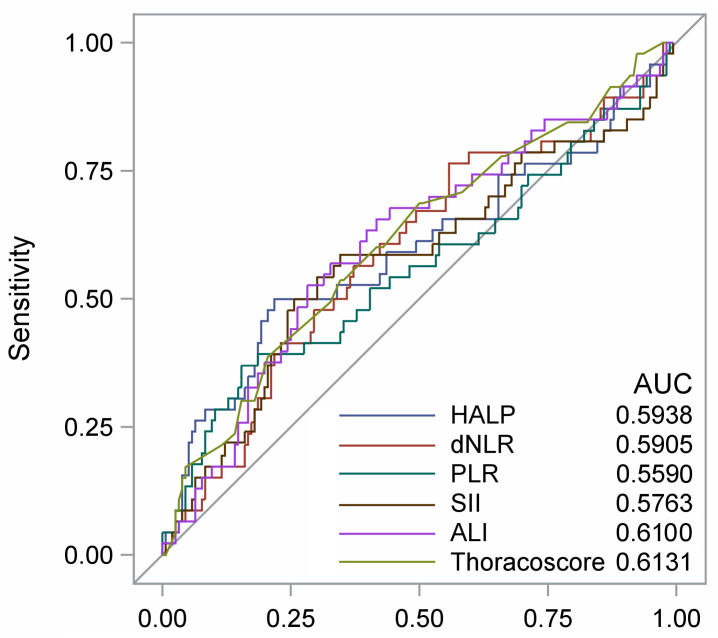
Receiver operating characteristic (ROC) curves for 3-year mortality of the Thoracoscore and of selected inflammatory parameters. HALP: hemoglobin, albumin, lymphocyte, and platelet score; NLR: derived neutrophil-to-lymphocyte ratio; PLR: platelet-to-lymphocyte ratio; SI: systemic immune-inflammation index; ALI: advanced lung cancer inflammation index. Best cut-off values (Youden index): HALP (32.2); NLR (2.29); PLR (196); SII (809); ALI (34.9); Thoracoscore (2.3).

**Figure 3 cancers-15-01854-f003:**
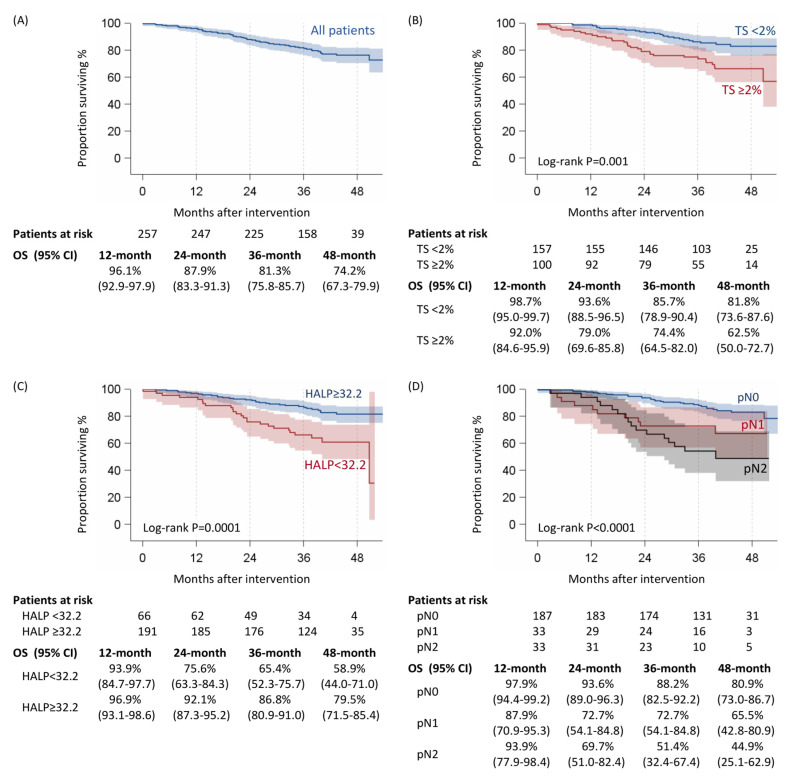
Overall survival of 257 patients who underwent lung cancer surgery overall (**A**), according to Thoracoscore (**B**), HALP (**C**), and pN (**D**).

**Table 1 cancers-15-01854-t001:** Components of the thoracic surgery scoring system (Thoracoscore) and their association with overall survival.

	Patients (%)	HR (95% CI)	*p* Value
All patients	257 (100)		
Age			
<55	32 (12.5)	1.00	
55–64	70 (27.2)	2.06 (0.59–7.23)	0.26
65+	155 (60.3)	2.90 (0.90–9.37)	0.08
Sex			
Males	149 (58.0)	1.00	
Females	108 (42.0)	0.91 (0.54–1.56)	0.74
Performance status			
0	162 (63.0)	1.00	
1	75 (29.2)	1.58 (0.90–2.78)	0.11
2	17 (6.6)	2.13 (0.88–5.14)	0.09
ASA score			
1	118 (45.9)	1.00	
2	86 (33.5)	1.58 (0.85–2.94)	0.15
3	48 (18.7)	1.72 (0.85–3.48)	0.13
Comorbidities			
None	11 (4.3)	1.00	
1	37 (14.4)	1.76 (0.39–7.95)	0.46
2	58 (22.6)	0.94 (0.21–4.31)	0.94
3	151 (58.8)	1.31 (0.31–5.47)	0.71
Any (1–3)	246 (95.7)	1.28 (0.31–5.27)	0.73
Cardiovascular	80 (31.1)	0.69 (0.37–1.29)	0.25
BPCO	99 (38.5)	1.09 (0.64–1.87)	0.73
Other malignancy	91 (35.4)	1.34 (0.79–2.29)	0.28
Dyspnea scale			
None	168 (65.4)	1.00	
1	56 (21.8)	1.22 (0.65–2.29)	0.54
2–4	18 (7.0)	2.12 (0.93–4.80)	0.07
Intervention			
Other †	238 (92.6)	1.00	
Pneumonectomy	19 (7.4)	2.99 (1.41–6.34)	0.004
Thoracoscore *			
<2%	155 (60.3)	1.00	
>2%	100 (38.9)	2.33 (1.37–3.95)	0.002

Data are missing for the following variables: PS (n = 3); ASA (n = 5); cardiovascular disease (n = 1); dyspnea (n = 15); * Thoracoscore (n = 2). † Lobectomy (n = 148); bilobectomy (n = 7); wedge resection (n = 8), segmentectomy (n = 75).

**Table 2 cancers-15-01854-t002:** Other factors and their association with overall survival.

	Patients (%)	HR (95% CI)	*p* Value
Body mass index (BMI)			
Underweight	13 (5.1)	1.58 (0.55–4.51)	0.39
Normal weight	121 (47.1)	1.00	
Overweight	84 (32.7)	0.85 (0.46–1.57)	0.60
Obese	39 (15.2)	0.89 (0.40–1.95)	0.76
Smoking status			
Non-smoker	41 (16.0)	1.00	
Ex-smoker	152 (59.1)	1.27 (0.56–2.88)	0.57
Current smoker	63 (24.5)	1.89 (0.79–4.53)	0.15
Symptoms			
No	193 (75.1)	1.00	
Yes	62 (24.1)	1.78 (1.03–3.07)	0.04
FEV1 %predicted			
≥80	172 (66.9)	1.00	
<80	84 (32.7)	1.67 (0.98–2.85)	0.06
FEV1/FVC (Tiffeneau)			
≥70	147 (57.2)	1.00	
<70	98 (38.1)	1.24 (0.72–2.12)	0.44
DLCO			
≥70	72 (28.0)	1.00	
<70	86 (33.5)	1.86 (0.98–3.52)	0.06
Histology			
ADK	188 (73.2)	1.00	
Epidermoid	54 (21.0)	2.25 (1.26–4.02)	0.006
Other NSCLC *	15 (5.8)	3.71 (1.63–8.44)	0.002
Stage			
I	145 (56.4)	1.00	
II	57 (22.2)	2.50 (1.28–4.91)	0.008
III	55 (21.4)	4.25 (2.27–7.95)	<0.0001
pN			
pN0	187 (72.8)	1.00	
pN1	33 (12.8)	2.41 (1.18–4.96)	0.02
pN2	33 (12.8)	4.17 (2.26–7.70)	<0.0001
Pleural invasion			
Pl0	164 (63.8)	1.00	
Pl+	93 (36.2)	2.78 (1.63–4.72)	0.0002
Perineural/vascular emboli			
No	102 (39.7)	1.00	
Yes	154 (59.9)	2.76 (1.62–4.73)	0.0002

FEV1: Forced-expiratory volume in 1 s; FEV1/FVC: Tiffeneau–Pinelli index; DLCO: diffusion capacity. Data are missing for the following variables: smoking (n = 1); symptoms (n = 2); FEV1 (n = 1); FEV1/FVC (n = 12); DLCO (n = 99); pN (n = 4); perineural invasion (n = 1). * 15 NSCLC (7 adeno-squamous, 4 sarcomatoid carcinomas, 4 undifferentiated carcinomas).

**Table 3 cancers-15-01854-t003:** Serological and inflammatory markers and their association with overall survival.

	Patients (%)	HR (95% CI)	*p* Value
SEROLOGICAL MARKERS			
CRP			
Normal (<3)	140 (54.5)	1.00	
High (≥3)	113 (44.0)	1.60 (0.94–2.71)	0.08
Albumin			
Low (<35)	12 (4.7)	1.87 (0.68–5.19)	0.23
Normal (≥35)	145 (56.4)	1.00	
Hemoglobin			
Low (<13)	88 (34.2)	1.97 (1.17–3.34)	0.01
Normal (≥13)	169 (65.8)	1.00	
Platelets			
Normal (<390)	243 (94.6)	1.00	
High (≥390)	14 (5.4)	2.19 (0.87–5.51)	0.09
Leucocytes			
Normal (<11)	231 (89.9)	1.00	
High (≥11)	26 (10.1)	1.17 (0.50–2.73)	0.72
Lymphocytes			
Normal (<3.8)	249 (96.9)	1.00	
High (≥3.8)	8 (3.1)	0.58 (0.08–4.16)	0.58
Neutrophils			
Normal (<6.8)	219 (85.2)	1.00	
High (≥6.8)	38 (14.8)	1.57 (0.81–3.04)	0.18
INFLAMMATORY MARKERS			
HALP			
<32.2	66 (25.7)	2.78 (1.64–4.72)	0.0002
≥32.2	91 (35.4)	1.00	
NLR			
<2.29	101 (39.3)	1.00	
≥2.29	156 (60.7)	2.14 (1.17–3.93)	0.01
PLR			
<196.1	209 (81.3)	1.00	
≥196.1	48 (18.7)	2.27 (1.28–4.02)	0.005
SII			
<808.9	175 (68.1)	1.00	
≥808.9	82 (31.9)	2.59 (1.53–4.38)	0.0004
ALI			
<34.86	85 (33.1)	2.55 (1.51–4.31)	0.0005
≥34.86	172 (66.9)	1.00	

Data for CRP are missing for 4 patients. CRP: c-reactive protein; HALP: hemoglobin, albumin, lymphocyte, and platelet score; NLR: derived neutrophil-to-lymphocyte ratio; PLR: platelet-to-lymphocyte ratio; SII: systemic immune-inflammation index; AL:, advanced lung cancer inflammation index. Cut-off values for serological markers are based on normal ranges, while those for inflammatory markers were obtained by receiver operating characteristic (ROC) curve analysis.

**Table 4 cancers-15-01854-t004:** Multivariate analysis.

Parameter	Values	HR (95% CI)	*p* Value
Thoracoscore	≥2% vs. <2%	1.92 (1.10–3.36)	0.02
Histology	Epidermoid vs. ADK	1.43 (0.73–2.79)	0.30
	Other NSCLC vs. ADK	3.57 (1.51–8.41)	0.004
Pathological N	pN1 vs. pN0	2.10 (0.98–4.48)	0.06
	pN2 vs. pN0	4.77 (2.53–8.98)	<0.0001
HALP	<32.2 vs. ≥32.2	2.30 (1.30–4.05)	0.004

ADK: adenocarcinoma; NSCLC: non-small-cell lung cancer; HALP: hemoglobin, albumin, lymphocyte, and platelet score. Hazard ratios (HRs) and 95% confidence intervals obtained from multivariate Cox proportional-hazards regression model with all variables fitted simultaneously.

**Table 5 cancers-15-01854-t005:** Correlation between inflammatory markers and tumor characteristics.

	pN				Stage			
	pN0	pN1	pN2	*p*-Value *	I	II	III	*p*-Value *
	N (%)	N (%)	N (%)		N (%)	N (%)		
HALP								
<32.16	44 (67.7)	13 (20.0)	8 (12.3)		27 (40.9)	20 (30.3)	19 (28.8)	
≥32.16	143 (76.1)	20 (10.6)	25 (13.3)	0.47	118 (61.8)	37 (19.4)	36 (18.8)	0.008
dNLR								
<2.67	75 (75.8)	12 (12.1)	12 (12.1)		63 (62.4)	21 (20.8)	17 (16.8)	
≥2.67	112 (72.7)	21 (13.6)	21 (13.6)	0.62	82 (52.6)	36 (23.1)	38 (24.4)	0.09
PLR								
<196.1	158 (76.7)	23 (11.2)	25 (12.1)		126 (60.3)	45 (21.5)	38 (18.2)	
≥196.1	29 (61.7)	10 (21.3)	8 (17.0)	0.08	19 (39.6)	12 (25.0)	17 (35.4)	0.004
SII								
<723.3	134 (77.9)	18 (10.5)	20 (11.6)		113 (64.6)	34 (19.4)	28 (16.0)	
≥723.3	53 (65.4)	15 (18.5)	13 (16.0)	0.08	32 (39.0)	23 (28.0)	27 (32.9)	<0.0001
ALI								
<34.86	57 (67.9)	14 (16.7)	13 (15.5)		38 (44.7)	24 (28.2)	23 (27.1)	
≥34.86	130 (76.9)	19 (11.2)	20 (11.8)	0.18	107 (62.2)	33 (19.2)	32 (18.6)	0.02

HALP: hemoglobin, albumin, lymphocyte, and platelet score; dNLR: derived neutrophil-to-lymphocyte ratio; PLR: platelet-to-lymphocyte ratio; SII, systemic immune-inflammation index; ALI: advanced lung cancer inflammation index. * Mantel–Haenszel test for trend.

## Data Availability

Not applicable.

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
