# Peer review of "Systemic Inflammation and Lung Cancer: Is It a Real Paradigm? Prognostic Value of Inflammatory Indexes in Patients with Resected Non-Small-Cell Lung Cancer"

_cancers, 2023, doi:10.3390/cancers15061854_

Round 1

Reviewer 1 Report

1. The inclusion criteria indicated patients with oligometastatic stage IV - brain and adrenal gland - (chemo or radiotherapy neo-adjuvant and adjuvant). Is this erroneously stated?

2. It seems to me that figure 3 is more appropriate after the table with multifactorial analysis.

3. The authors did not calculate correlations between parameters: for example, does the level of inflammatory markers correlate with the status of lymph node involvement or disease stage?

4. Is it possible to form groups for which several factors from Table 4 will be unfavorable / favorable at the same time? How will the hazard ratio change then?

Author Response

Question n. 1

The inclusion criteria indicated patients with oligometastatic stage IV - brain and adrenal gland - (chemo or radiotherapy neo-adjuvant and adjuvant). Is this erroneously stated?

It was erroneusly stated. We deleted the sentence. Thank you.

Question n. 2.

It seems to me that figure 3 is more appropriate after the table with multifactorial analysis.

Done

Question n. 3.

The authors did not calculate correlations between parameters: for example, does the level of inflammatory markers correlate with the status of lymph node involvement or disease stage?

We added this aspect into the text “On the basis of univariate and multivariate analysis, we calculate correlations between the level of inflammatory markers and  the status of lymph node involvement or disease stage. HALP (p: 0.008), PLR (0.004), SII (<0.0001) and ALI (0.02) are associated with more invasive stage. (table n. 5)” and we add table n. 5;

Question n. 4

Is it possible to form groups for which several factors from Table 4 will be unfavorable / favorable at the same time? How will the hazard ratio change then?

I did not quite understand the issue that the reviewer brings to our attention. At univariate and multivarate analysis you can clearly see which factors are favourable/unfavourable and impacting on survival.

Reviewer 2 Report

This is a well written manuscript looking into various inflammatory markers as prognostic factors for resected NSCLC lung cancer. 

Major comment: The authors describes their MVA was done in a stepwise mode but they do not show how they selected variables in the model. While several clinical and inflammatory markers were significant on UVA, they are not shown/ included in their final MVA model to reflect that they are not significant. It is not clear whether the above is true or they fell out of the model. Please provide that information clearly.

Thoracosore is something that is commonly used in thoracic surgery but may not be familiar to broad readers. It will be important/ helpful to provide/ describe how it is calculated. 

The KM figures has different colors but does not have text showing which color represent which group unless the reader is looking at the number of of patients at risk carefully. Please add text against each figure so it is easy to read.

Please provide complete words for each abbreviation used. eg. NIV on page 4, line 181

Some English grammar correction needed.  eg. Patients undergoing to lung surgery on page 1, line 22

Author Response

This is a well written manuscript looking into various inflammatory markers as prognostic factors for resected NSCLC lung cancer. 

Major comment:

Question 1

The authors describes their MVA was done in a stepwise mode but they do not show how they selected variables in the model. While several clinical and inflammatory markers were significant on UVA, they are not shown/ included in their final MVA model to reflect that they are not significant. It is not clear whether the above is true or they fell out of the model. Please provide that information clearly.

We introduced into the MVA model the only significative variables at univariate analsysis. All the variables non-significative into MVA model, were not reported in the MVA table.

Question 2

Thoracosore is something that is commonly used in thoracic surgery but may not be familiar to broad readers. It will be important/ helpful to provide/ describe how it is calculated. 

We added these informations in the text

“For all patients we calculated Thoracoscore at the time of hospitalization (25) (table n.1). Thoracoscore was described in 2007 as model for risk of in-hospital death among adult patients after general thoracic surgery. It uses only 9 pre-operative variables (age, sex, ASA classification, performance status, dyspnea score, priority of surgery, procedure, diagnosis, comorbidity) and it is recognized as a valid clinical tool for predicting the risk of death. The thoracoscore evaluation is easily calculated and available online (https://sfar.org/scores2/thoracoscore2.php).

Question n. 3

The KM figures has different colors but does not have text showing which color represent which group unless the reader is looking at the number of of patients at risk carefully. Please add text against each figure so it is easy to read.

We changed the KM figures.as requested.

Question n. 4

Please provide complete words for each abbreviation used. eg. NIV on page 4, line 181

Done

Question n. 5

Some English grammar correction needed.  eg. Patients undergoing to lung surgery on page 1, line 22

We performed a formal english revision of the paper.

Reviewer 3 Report

ABSTRACT OK,

MATERIALS OK

RESULTS OK 

DISCUSSION OK

CONCLUSIONS OK

THE TEXT NEED A "RETOUCH" IN ENGLISH LANGUAGE

I THINK THE IDEA OF THE AUTHORS WAS GOOD AND IS EXPRESSED IN THE TEXT.BUT THE RESULTS COULD BE CONSIDERED AS THE FIRST STEP FOR FURTHER INVESTIGATION . 

Author Response

REVIEWER 3

BSTRACT OK,

MATERIALS OK

RESULTS OK 

DISCUSSION OK

CONCLUSIONS OK

Question n. 1

THE TEXT NEED A "RETOUCH" IN ENGLISH LANGUAGE

Done

Question n. 2

I THINK THE IDEA OF THE AUTHORS WAS GOOD AND IS EXPRESSED IN THE TEXT.BUT THE RESULTS COULD BE CONSIDERED AS THE FIRST STEP FOR FURTHER INVESTIGATION . 

We talk about the necessity of other studies in the conclusions.

Round 2

Reviewer 1 Report

1. We kindly ask you to add information on how the threshold values for all Indexes of Inflammatory Status are chosen. Line 156 simply specifies the cutoff values, but does not explain the choice.

2. Regarding table 4, I meant that, for example, pN2 and HALP<32.2 are factors for poor prognosis. If we consider a group of patients in whom both of these factors are unfavorable) compared to a group of pN0 and HALP>32.2 (favorable factors), then what will be the prognosis?

Author Response

1. We kindly ask you to add information on how the threshold values for all Indexes of Inflammatory Status are chosen. Line 156 simply specifies the cutoff values, but does not explain the choice.

As described in the methods section and footnote of figure 2 the cut-off was determined by receiver operating curves (ROC) analysis: “Optimal cut-off for continuous biological variables were determined by the Youden-J using receiver operating curves (ROC) analysis (figure n.2). Different cut-offs are shown in the table and precisely: HALP 32.2; NLR 0.01; PLR 196.1; SII 808.9; ALI 34.86.” This is a standard statistical method to choose a cut-off which maximize sensitivity and specificity of a marker.

2.     Regarding table 4, I meant that, for example, pN2 and HALP<32.2 are factors for poor prognosis. If we consider a group of patients in whom both of these factors are unfavorable) compared to a group of pN0 and HALP>32.2 (favorable factors), then what will be the prognosis?

Compared to a group of pN0 and HALP>32.2, the hazard of dying for those pN2 and HALP<32.2 is 4.77 (HR for pN2 vs pN0)  x 2.30(HR <32.2 vs ≥32.2) =11 fold . This is standard reading of multivariate analysis coefficients. For the reviewers interest, in our study, 7 of the 8 patients with such characteristics (pN2 and HALP<32.2) died, vs 18 of 143 patients pN0 and HALP>32.2.

Reviewer 3 Report

OK

Author Response

Thank you